# Quantification of Arbutin in Cosmetics, Drugs and Food Supplements by Hydrophilic-Interaction Chromatography

**DOI:** 10.3390/molecules27175673

**Published:** 2022-09-02

**Authors:** Sarah Repert, Sandra Matthes, Wilfried Rozhon

**Affiliations:** Department of Agriculture, Ecotrophology, and Landscape Development, Anhalt University of Applied Sciences, 06406 Bernburg, Germany

**Keywords:** arbutin, cosmetics, drugs, food supplements, HPLC, hydrophilic-interaction chromatography

## Abstract

Arbutin, the glucoside of hydroquinone, exists in two isomers, α-arbutin and β-arbutin. The synthetic α isomer is mainly used as a skin brightening agent, while β-arbutin occurs naturally, for instance in bearberry, and is used in drugs for treatment of lower urinary tract infections and as a food supplement. Since both isomers can be harmful at high concentrations, methods for their quantification are required. Classically they have been determined by reversed-phase chromatography, but separation of both isomers is often unsatisfactory. Here we present a simple and reliable method for quantification of α- and β-arbutin based on hydrophilic-interaction chromatography. Prior to analysis, interfering compounds that would frequently be present in cosmetics and drugs, particularly biopolymers, were efficiently removed by precipitation with acetonitrile. In this paper, for separation, a Cyclobond I 2000 5 µm 250 × 4.6 mm column was employed as stationary phase and acetonitrile/water 92/8 (*v/v*) was used as an eluent at a flow rate of 0.8 mL min^−1^. For quantification, a UV detector operating at 284 nm was applied. Although analysis took less than 10 min, baseline separation of α- and β-arbutin was achieved. The response was highly linear (r > 0.999) and the method had, for both α- and β-arbutin, a LOD of 0.003% (*w*/*w*) and a LOQ of 0.009% (*w*/*w*). Moreover, the method showed excellent intra-day and inter-day repeatability with relative standard deviations in the range of 0.5% to 2.3% and 1.0% to 2.2%, respectively, with cosmetics, drugs and food supplements as samples.

## 1. Introduction

Skin whitening is a widespread trend, especially in Japan, India and Africa. In East Asia, light skin is often seen as a symbol of wealth and many people associate it with social privileges such as better jobs or marital prospects [1]. Due to these and other motivating aspects, for instance beauty ideals, or due to a hyperpigmentary disorder, many people use skin lightening products. Such cosmetics contain agents that inhibit the copper-containing enzyme tyrosinase, which is responsible for the synthesis of the colouring pigment melanin. Melanin is produced in the melanocytes and an over-production leads to hyperpigmentation of the human skin [2]. There are several agents for depigmentation and skin lightening, for instance arbutin, hydroquinone, catechins, kojic acid, tranexamic acid and azelaic acid [3]. Mercury salts, corticosteroids and glycolic acid have also been used for this purpose [1].

Arbutin is a tyrosinase inhibitor which exists in two configurations: 4-hydroxyphenyl-α-D-glucopyranoside (α-arbutin) and 4-hydroxyphenyl-β-D- glucopyranoside (β-arbutin). α-Arbutin is a synthetic compound, while the latter isomer occurs naturally in various plant families, for instance in Ericaceae, Asteraceae and Rosaceae [4]. Under acidic conditions, both isomers hydrolyse to D-glucose and hydroquinone [5]. The mechanism of action of arbutin is based on competitive inhibition of the enzyme tyrosinase [2]. Therefore, arbutin competes with hydroxylation of the amino acid tyrosine and thereby prevents formation of L-DOPA. As a result, oxidation of L-DOPA to L-dopaquinone cannot takes place and melanin biosynthesis is inhibited [5,6]. Using B16F10 mouse melanoma cells, Jun et al. showed that β-arbutin inhibits melanin synthesis efficiently at concentrations of 5 mmol L^−1^ or more. In addition, using an enzymatic assay, they showed that the half maximal inhibitory concentration (IC_50_) of β-arbutin against mushroom tyrosinase was 6 mmol L^−1^. In this assay, tyrosine was used as a substrate and the enzymatic activity was quantified via measurements of the absorbance at 475 nm [7].

As stated above, hydrolysis of arbutin leads to hydroquinone, a potentially harmful and controversially discussed compound. The use of hydroquinone itself in cosmetics is prohibited in the European Union [8,9] since it shows cytotoxicity to melanocytes. Side effects of long-term use may be exogenous onchronosis, leucoderma and even carcinogenesis [10]. In contrast, arbutin causes much less or no irritation to the skin [11,12]. An upper limit of content of arbutin in cosmetics of 2% (*w/w*) for the α-isomer and 7% (*w/w*) for β-arbutin has been accepted until recently in the European Community [13]. However, in a very recent opinion, the Scientific Committee on Consumer Safety (SCCS) of the European Commission refused declaring those concentrations safe and concluded that too few data are available to allow establishing safe concentration limits for α-isomer and β-arbutin [11].

Leaf extracts of bearberry (*Arctostaphylos uva-ursi*), often referred to as uva ursi extracts, are rich in β-arbutin and are frequently used for treatment of lower urinary tract infections [14]. The European Medicines Agency (EMA) concluded that, based on its traditional use, the effectiveness of uva ursi extracts was plausible and that its use was safe [15]. However, the EMA noted that clinical studies allowing final assessment of the potency and safety of β-arbutin were still lacking. After oral administration, β-arbutin is absorbed in the small intestine and hydrolysed in the liver by β-glucosidases to hydroquinone, which is subsequently conjugated with glucuronic acid or sulfuric acid. In these forms it is excreted by the kidney. In case of a lower urinary tract infection, the conjugates are hydrolysed in the urinary bladder [16] and the formed hydroquinone shows antibacterial effects [17]. While β-arbutin is sometimes also used as skin brightener [18], α-arbutin has not been used for treatment of lower urinary tract infections.

Several methods for quantification of arbutin based on spectrophotometry have been described. Arbutin reacts with iron(III) chloride to a complex, which was measured at 292 nm [19]. An indirect spectrophotometry technique made used of the reaction of arbutin with periodate. Excess periodate causes liberation of iodine, which was detected at 351 nm [19]. However, reducing compounds interfered with this method. In an alternative spectrophotometric approach, arbutin was reacted with 4-aminoantipyrine in an alkaline medium in the presence of hexacyanoferrate(III) to a red-coloured product that was measured at 514 nm [4]. In a recently described method, arbutin was quantified by fluorimetry after oxidation of the phenolic residue by treatment with hypochlorite to a benzoquinone [20]. Another highly sensitive method for quantification of arbutin was based on amperometry and used a nano sepiolite-clay carbon paste electrode [21]. A clear drawback of these methods was their limited specificity and inability to discriminate between the two arbutin isomers. Thus, a number of separation techniques including thin-layer chromatography [5], capillary zone electrophoresis [22], micellar electrokinetic capillary chromatography with amperometric detection [23], and gas chromatography [24] have been developed for quantification of arbutin in medicinal plants, drugs and cosmetics. However, the most frequently used technique is high performance liquid chromatography (HPLC) in the reversed-phase mode using C18 columns [25,26,27,28,29,30,31,32]. Usually, UV detection at different wavelengths, e.g., at 280 nm [28], 289 nm [31] or 254 nm [27], is used. While separation of arbutin from the less-polar hydroquinone is easily achievable, the retention times of α- and β-arbutin are very similar in reversed-phase chromatography and the peaks frequently partially overlap [22]. Moreover, many drugs and cosmetics contain biopolymers, particularly hyaluronic acid, that must be removed prior to analysis. A general and simple procedure for biopolymer removal is precipitation with organic solvents. However, the presence of a high concentration of organic solvents in a sample is unfavourable for reversed phase chromatography of highly polar compounds, because injection of a sample dissolved in a solvent with a much higher elution strength than the mobile phase causes peak broadening [33]. In addition, lipids, frequently present in cosmetics, increase the analysis time considerably because high concentrations of methanol or acetonitrile are required to remove such compounds from C_18_ columns. In sharp contrast, high contents of organic solvents are ideal for hydrophilic-interaction chromatography [34] and, moreover, lipophilic compounds do not interfere because they elute first.

Here, we describe a simple and efficient method for quantification of arbutin by hydrophilic-interaction chromatography in combination with UV-detection. Biopolymers are removed by addition of acetonitrile, subsequent centrifugation and membrane filtration. Samples prepared in this way can be directly injected into an HPLC system equipped with a Cyclobond I 2000 column. Importantly, the peaks of α- and β-arbutin are baseline separated, and the method is characterised by a high recovery rate and excellent reproducibility.

## 2. Results and Discussion

### 2.1. Spectra of α-Abrutin and β-Arbutin

α-Arbutin and β-arbutin (Figure 1A) contain phenolic hydroxy groups that can be deprotonated at a high pH, which might impact on the UV spectra. To investigate this in more detail, the UV spectra of both isomers were recorded in the range of 200 to 400 nm at pH values of 2, 7 and 12 (Figure 1B,C). Both compounds showed almost identical spectra. The spectra of each compound were, at least at wavelength exceeding 215 nm, hardly different at pH 2 and pH 7. Under these conditions, the absorption maxima were at 221 nm and 284 nm, with a minimum at 245 nm. In contrast, at pH 12, where the phenolic groups were deprotonated, the first absorption maxima shifted to 234 nm and the second to 300 nm, with a minimum in between, at 267 nm. These data indicated that at pH values of eluents compatible with the columns used here (pH 2 to 7), both arbutin isomers were protonated and, therefore, uncharged, which is the favourable form for hydrophilic-interaction chromatography.

To confirm the identity of the compounds used in this study, we also recorded their optical rotation at different wavelengths (Figure 1D). The measured specific rotations at 589 nm, of +180.0° and −66.3°, fitted perfectly to values reported in the literature for α-arbutin and β-arbutin, respectively [8,35].

Since the eluents used in this study have a pH close to 7, we used a wavelength of 284 nm for detection. Although a wavelength of 221 nm would allow more sensitive detection, 284 nm was selected since less interference was expected at a higher wavelength. In addition, high sensitivity is not an important issue for determination of arbutin in cosmetic products and drugs since both contain α-arbutin and β-arbutin in the % range.

### 2.2. Optimisation of Chromatographic Conditions

In hydrophilic-interaction chromatography, polar phases are frequently used in combination with ACN/water or methanol/water mixtures. Here, we tested Cyclobond I 2000, Nucleodex beta-OH, Nucleodur NH2, and Lichrospher CN as stationary phases, in combination with ACN/water as the mobile phase (preliminary experiments had shown that methanol/water mixtures were not suitable due to too low retention of the compounds).

Preliminary experiments showed that both arbutin isomers were insufficiently retained on the CN phase even at high ACN concentrations. In contrast, sufficient retention was obtained with the three other phases. To study separation on these phases in more detail, separation of α- and β-arbutin was investigated in the isocratic mode with eluents consisting of ACN/water 80/20 (*v*/*v*) to ACN/water 98/2 (*v*/*v*). However, with Nucleodex beta-OH as a stationary phase, only very broad peaks were obtained. In contrast, with Cyclobond and Nucledour NH2 phases, well separated and sharp peaks were obtained with eluents containing ACN/water 92/8 (*v*/*v*) or ACN/water 94/6 (*v*/*v*) ACN, respectively (Figure 2A–C). To study separation on these phases in more detail, the capacity factors (k′) of α- and β-arbutin were determined and plotted against the ACN content (Figure 2D,E). This showed a clear increase in k′ with the ACN content for both columns. In general, the separation was better with the Cyclobond I 2000 column than with the Nucleodur NH2 column, as indicated by the more different k′ values for α- and β-arbutin and the higher resolution. Interestingly, the resolution of both compounds increased, with the ACN content for the Cyclobond I 2000 column up to ACN/water 96/4 (*v*/*v*) (Figure 2D). A further increase in the ACN content enhanced resolution only slightly, while the capacity factors were strongly increased. This contradicting result could be explained by the tremendously increased peak asymmetry at very high ACN contents (Figure 2B,D). Taking these results together, the most preferable chromatographic conditions were obtained using a Cyclobond I 2000 column in combination with ACN/water 92/8 (*v*/*v*) as eluent, which allowed baseline separation of both arbutin isomers in less than 10 min.

Having established the most suitable eluent composition, we next investigated the impact of the flow rate on the theoretical plate height of α- and β-arbutin. Plotting the data in a Van Deemter diagram [36] showed a minimum for the theoretical plate height and best resolution at a flow rate of 0.4 mL min^−1^ (Figure 3). At lower flow rates the theoretical plate height increased rapidly. At higher flow rates, the theoretical plate height increased and peak resolution dropped, but the separation was quicker. A flow rate of 0.8 mL min^−1^ was selected for further experiments since this allowed good separation in a reasonable time.

### 2.3. Sample Preparation

Cosmetic products contain frequently biopolymers, particularly hyaluronic acid. Their removal is possible with high concentrations of organic solvents. However, loss of analytes may happen by co-precipitation or binding to the precipitated polymer. Thus, we used SPN-5 for spiking experiments. SPN-5 is a cosmetic serum representing a matrix similar to skin brighteners but free of any arbutin and, thus, a suitable matrix for spiking experiments. A-Arbutin or β-arbutin were added to SPN-5 to a final content of exactly 2% (*w*/*w*). As controls, solutions of the compounds were prepared in water at the same content. Next, acetonitrile was added to a final content of ACN/water 80/20 (*v*/*v*) or ACN/water 92/8 (*v*/*v*), the solution mixed well, centrifuged, filtered through a 0.22 µm membrane filter and analysed using the previously established conditions. A similar experiment was also performed using 2-propanol (2-PrOH) as solvent for precipitation (final content: 2-PrOH/water 80/20 (*v*/*v*)).

As indicated in Table 1, the recovery rates of both arbutin isomers were close to 100%, confirming that no sample loss occurred during sample preparation. However, presence of 2-PrOH in the sample injected into the HPLC system distorted peak shape (Figure 4) and clearly increased the peak width at half peak height (W_1/2_) (Table 1). This might have resulted from the much higher viscosity of 2-PrOH compared with ACN [33] and from the presence of hydroxy groups in this solvent, which might interfere with the binding of arbutin to the hydroxy groups of the β-cyclodextrin stationary phase. In contrast, neither ACN/water 80/20 (*v*/*v*) nor ACN/water 92/8 (*v*/*v*) had such an effect. Since samples with ACN/water 92/8 (*v*/*v*) showed the smallest W_1/2_, this method of sample preparation was selected for further experiments.

### 2.4. Validation of the Method

α-Arbutin and β-arbutin showed an almost identical detector response at 284 nm and the calibration curves were perfectly linear up to 1000 mg L^−1^ (Figure 5).

The lower limit of detection (LOD) and the lower limit of quantification (LOQ) were determined according to ICH guideline Q2(R2) on Validation of Analytical Procedures [37]. Using the calibration curve, the LOQ for α- and β-arbutin were roughly estimated to be in the range of 2 mg L^−1^, corresponding to a LOQ of 0.01% (*w*/*w*) in samples prior to precipitation with ACN if a sample amount of 100 mg was used. Subsequently, SPN-5 was spiked with both compounds to a level of 0.01% (*w*/*w*). Analysis of 20 aliquots revealed that the standard deviation of the solution for both compounds was 0.18 mg L^−1^, corresponding to 0.0009% (*w*/*w*) in the original sample. Accordingly, the LOD and LOQ for the dissolved sample were 0.6 mg L^−1^ and 1.8 mg L^−1^, respectively. This corresponded to a LOD of 0.003% (*w*/*w*) and a LOQ of 0.009% (*w*/*w*) if a sample of 100 mg was used for analysis (Appendix A).

To assess the recovery rate of α-arbutin, two cosmetic sera, The Ordinary Alpha Arbutin 2% + HA, and The Ordinary Ascorbic Acid 8% + Alpha Arbutin 2%, were analysed before and after spiking with 2% (*w*/*w*) α-arbutin. The recovery rate of β-arbutin was assessed by spiking a solution of UROinfekt, a drug for treating lower urinary tract infections, as well as a solution of a food supplement containing bearberry extract. Analysis in quadruplicate indicated recovery rates close to 100% for all matrices tested (Table 2).

To assess reproducibility of the method, the two cosmetic sera, the drug and the food supplement mentioned above, were aliquoted and analysed in quadruplicate on four consecutive days (Table 3, Table 4, Table 5 and Table 6). The results showed that the α-arbutin contents of both sera, 1.91 ± 0.02% (*w*/*w*) and 1.89 ± 0.03% (*w*/*w*), were close to the declared values of 2% (*w*/*w*). For the cosmetic sera, the results for the individual days (intra-day) showed relative standard deviations (RSD) in the range of 0.5% to 2.3% for α-arbutin. The overall (inter-day) result showed RSDs of 1.0% and 1.5% for α-arbutin. β-Arbutin was below the detection limit in both sera. The drug and the food supplement contained only β-arbutin. The intra-day RSDs for the determination of β-arbutin in these products ranged from 0.8 to 1.8%, and the inter-day RSDs were 2.2% and 1.0% for UROinfekt and the bearberry food supplement, respectively. For the food supplement, the found content of 103.8 ± 1.0 mg per pill was very close to the declaration of 100 mg per pill. In contrast, for the drug, a deviation was found: the measured content of 159.5 ± 3.8 mg per pill was below the declaration of 180 to 210 mg per pill. However, according to the declaration, the arbutin content of the product was determined by spectrophotometry and, thus, might be slightly overestimated, since other plant metabolites were also present in the extract which absorb at 284 nm, as indicated in Figure 6D,H.

The calibration curves recorded on the individual days were highly similar (Table 7) and all Pearson correlation coefficients exceeded 0.9997.

To assess peak purity, UV spectra were recorded using a diode array detector and compared with authentic standards. As indicated in Figure 6, the UV spectra of the α- and β-arbutin peaks of the samples were identical to those of the authentic references, confirming purity of the peaks. Thus, the chromatographic conditions were suitable for specific detection of α- and β-arbutin in samples including cosmetics, drugs and food supplements. Using a DAD detector is particularly recommended for matrices with strong interference. This allows verification of peak purity since contaminating compounds usually have other UV spectra than the analyte, as indicated in Figure 6H.

### 2.5. Assessing the Method under the Principles of Green Chemistry

To assess the developed method under the principles of green chemistry we used the AGREE—Analytical GREEnness Metric Approach [38]. Using the AGREE calculator, a score of 0.66 was obtained, which was quite good compared with the examples for HPLC-based methods given in Pena-Pereira et al., 2020 [38]. According to the AGREE report (Appendix A), the weakest points of the method were principle 3 (preference of in situ measurements) and principle 10 (use of renewable reagents). For principle 3 improvement is hardly possible for a HPLC-based method applied to analysis of cosmetics and drugs: the method is necessarily offline. With respect to principle 10, it is noted that most solvents used in HPLC are not renewable and, thus, a bad rating in this category was expected. However, for most other principles a very good rating was obtained. For instance, only minute amounts of samples were required (principle 2), sample preparation was simple (principle 4), little energy was needed (principle 9; according to our estimates, approximately 0.05 kWh per sample), and no derivatisation (principle 6) or toxic reagents (principle 11) were required. Thus, in summary, the method achieved a good rating under the principles of green chemistry.

## 3. Materials and Methods

### 3.1. Reagents

α-Arbutin and β-arbutin had a purity of >99% and were purchased from Pure Health Solutions (Marrickville, Australia). Acetonitrile (ACN) and 2-propanol (2-PrOH) were of HPLC gradient grade and obtained from VWR (Radnor, PA, USA). Hydrochloric acid, phosphoric acid, sodium hydroxide and chloroform were of analytical grade and purchased from Merck Millipore (Burlington, MA, USA). Water used had less than 0.06 µS cm^−1^ and was purified with a Synergy UV system (Merck Millipore, Burlington, MA, USA).

### 3.2. UV Spectroscopy and Measurement of Optical Rotation

For UV spectroscopy, α-arbutin and β-arbutin were dissolved in 10 mM hydrochloric acid (pH 2), 10 mM sodium phosphate buffer (pH 7) or 10 mM sodium hydroxide (pH 12) to a concentration of 50 mg L^−1^. UV spectra were recorded using a Specord plus 200 photometer (Analytik Jena, Jena, Germany) from 200 to 400 nm with 1 nm datapoints and a slit of 2 nm. Cuvettes with an optical path length of 10 mm made of quartz glass of the grade high performance (Hellma, Müllheim, Germany) were used.

Optical rotation was measured with a 341 polarimeter (Perkin Elmer, Waltham, MA, USA) using a 10 cm cuvette tempered to 20 °C, and solutions of the compounds in water at concentrations of 0.01 g mL^−1^.

### 3.3. Sample Preparation and HPLC

Liquid samples were weighed (depending on the application, 30 mg or 100 mg) into 5 mL volumetric flasks and the weight noted to the nearest 0.1 mg. Ultrapure water (<0.06 µS cm^−1^), purified with a Synergy UV system (Merck Millipore, Burlington, MA, USA) was added to a total volume of 0.4 mL (for ACN/water 92/8 (*v*/*v*)) or 1 mL (for ACN/water or 2-PrOH/water 80/20 (*v*/*v*)). Then the flask was filled with ACN or 2-PrOH to the mark, and the content was mixed well. An aliquot of approximately 1.5 mL was centrifuged using a Biofuge pico (Heraeus, Hanau, Germany) at 13,000× *g* for 10 min. The supernatant was passed through a 0.22 µm nylon syringe filter of 13 mm diameter (BGB Analytik, Boeckten, Switzerland) directly into the HPLC vials.

### 3.4. HPLC

For initial experiments (optimisation of the HPLC conditions), an LC-10 HPLC system (Shimadzu, Nakagyo-ku, Japan) configured as described previously [39] was used. The column oven was operated at 25 °C and the UV detector was set to 284 nm. Columns tested included a Cyclobond I 2000 5 µm 250 × 4.6 mm column (Sigma-Aldrich, St. Louis, MO, USA), a Lichrospher 100 CN 5 µm 250 × 4 mm column (Merck Millipore, Burlington, MA, USA), a Nucleodur 100-5 NH2 5 µm 125 × 4.6 mm column and a Nucleodex beta-OH 5 µm 200 × 4 mm column (both from Macherey-Nagel, Düren, Germany). A flow rate of 0.8 ml L^−1^ was used. The eluents (isocratic elution) were formed using a gradient mixer by mixing eluent A (ACN/water 80/20 (*v*/*v*)) and eluent B (pure ACN) at the required ratios. For subsequent experiments, a series 1260 Infinity HPLC system (Agilent, Santa Clara, CA, USA) equipped with a 1260 VL quaternary pump, a 1260 ALS autosampler, a 1260 TCC column oven (operated at 25 °C) and either a 1260 MWD VL UV/VIS detector (set to 284 nm) or a 1260 DAD VL diode array detector, was used. For the 1260 system, pre-mixed ACN/water 92/8 (*v*/*v*) was used as eluent in the isocratic mode. Chloroform was found to not be retained on the Cyclobond I 2000 stationary phase. Thus, this compound was used to determine t_0_, the dead volume of the system, which was required for calculation of the capacity factors (k′).

### 3.5. Validation

The LOD and LOQ were determined according to The ICH guideline Q2(R2) on validation of analytical procedures [37] and defined as the 3.3-fold SD and 10-fold SD, respectively. The SD was estimated by spiking a blank matrix with the compounds to a level of the estimated LOQ. For these spiking experiments SPN-5 (SRS International, Brussels, Belgium) was used as a blank matrix, a cosmetic serum that does not contain arbutin. The same blank matrix was also used for the investigation of the optimal solvent for precipitation of biopolymers before analysis. Therefore, SPN-5 was spiked with α- or β-arbutin to 2% (*w*/*w*) and the recovery rates were determined.

The recovery rate of α-arbutin was measured by spiking the skin brighteners The Ordinary Alpha Arbutin 2% + HA (hyaluronic acid) and The Ordinary Ascorbic Acid 8% + Alpha Arbutin 2% (The Abnormal Beauty Company, New York City, NY, USA) with 2% (*w*/*w*) α-arbutin and measuring the contents before and after spiking. For measuring the recovery rate of β-arbutin the drug UROinfekt (Omega Pharma Deutschland GmbH, Herrenberg, Germany, declared β-arbutin content: 180–210 mg per pill) was dissolved in distilled water to a total volume of 10 mL and the solution measured before and after spiking with β-arbutin at a level of 20 g L^−1^ (corresponding to 200 mg arbutin per pill). The same experiment was also performed with bearberry extract food supplement pills (Warnke Vitalstoffe, Wetzlar, Germany, declared β-arbutin content: 100 mg per pill).

For intra-day and inter-day reproducibility the same skin whiteners as used before were analysed four times on four consecutive days. For each analysis approximately 30 mg sample were weighed to the nearest 0.1 mg, water added to 0.4 mL and ACN added to a total volume of exactly 5 mL. In addition, a pill of UROinfekt was dissolved in water to a total volume of 11 mL and the solution analysed in the same way as described above. Bearberry extract food supplement pills were analysed in the same way except that the pill was dissolved in water to a total volume of 5.4 mL.

For regression analysis linear fit with 1/x weighing was used. Outliers were identified using the Grubbs test.

## 4. Conclusions

This study showed that hydrophilic-interaction chromatography allows reliable analysis of the arbutin isomers in cosmetics, drugs, and food supplements. The method showed high intra-day and inter-day repeatability with relative standard deviations of less than 2.5%. The method had a high sensitivity as indicated by a LOD of 0.003% (*w*/*w*) and a LOQ of 0.009% (*w*/*w*) and is, therefore, also suitable for the detection of small arbutin quantities. Importantly, the method allows baseline separation of both isomers in less than 10 min, an important advantage compared with reversed-phase chromatography, where frequent overlapping of the α- and β-arbutin peaks has been observed [29], or a very long separation time is required [32]. Another advantage is that removal of biopolymers by precipitation with ACN is perfectly suitable with hydrophilic-interaction chromatography since high contents of organic solvents in the injected solution are compatible with this technique. In the samples analysed, no interference from the matrix or other active ingredients was observed. In addition, sample preparation was simple, use of toxic chemicals and derivatisation could be avoided, and very little sample was required. Thus, the presented method is time saving, and allows rapid and reproducible quantification of both arbutin isomers in a variety of sample types.

## Figures and Tables

**Figure 1 molecules-27-05673-f001:**
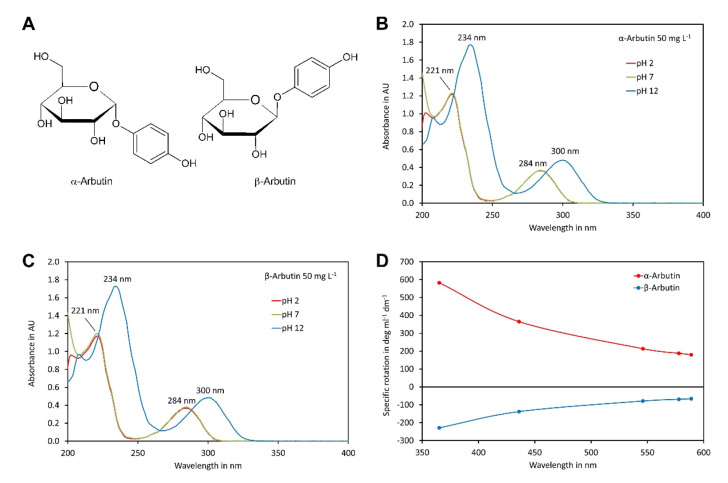
Spectra of α-arbutin and β-arbutin: (**A**) structures of the isomers; (**B**) UV spectra of α-arbutin at a concentration of 50 mg L^−1^ at pH 2, 7 and 12; (**C**) same as B but for β-arbutin; (**D**) ORD curves for α- and β-arbutin dissolved in water. The original data are available in Appendix A.

**Figure 2 molecules-27-05673-f002:**
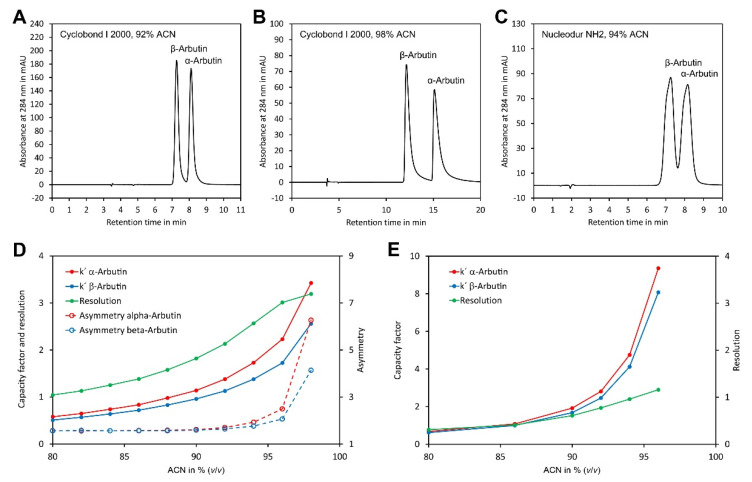
Optimisation of the chromatographic conditions for separation of α- and β-arbutin: (**A**) separation of α- and β-arbutin using a Cyclobond I 2000 5 µm 250 × 4.6 mm column and ACN/water 92/8 (*v*/*v*) as eluent; (**B**) as in A but using ACN/water 98/2 (*v*/*v*) as eluent; (**C**) as in A but using a Nucleodur NH2 5 µm 125 × 4.6 mm column and ACN/water 94/6 (*v*/*v*) as eluent; (**D**) dependency of the capacity factors, resolution and peak asymmetry of α- and β-arbutin on a Cyclodex I 2000 column from the ACN concentration; (**E**) dependency of the capacity factors and resolution of α- and β-arbutin on a Nucleodur NH2 column as stationary phase. The original data are available in Appendix A.

**Figure 3 molecules-27-05673-f003:**
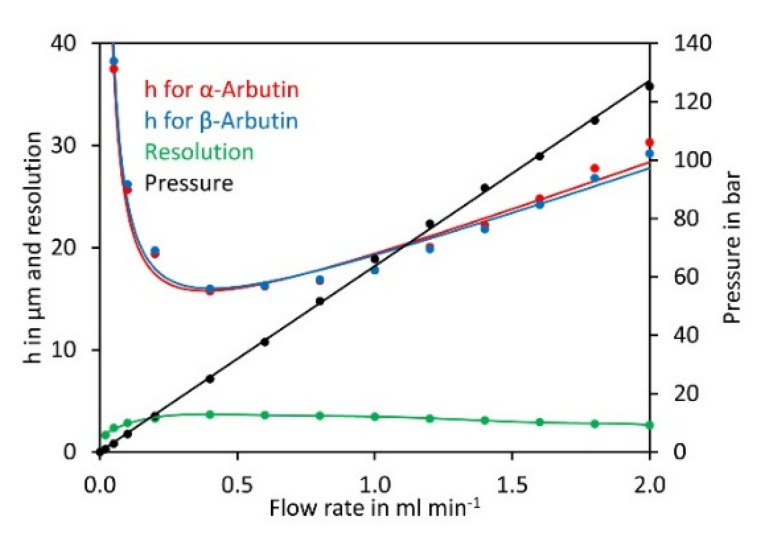
Van Deemter diagram for α-arbutin and β-arbutin. Both compounds were separated on a Cyclobond I 2000 5 µm 250 × 4.6 mm column using ACN/water 92/8 (*v*/*v*) at flow rates from 0.05 to 2 mL min^−1^. For convenience, the flow rate was plotted on the x-axis rather than the velocity of the mobile phase. The original data are available in Appendix A.

**Figure 4 molecules-27-05673-f004:**
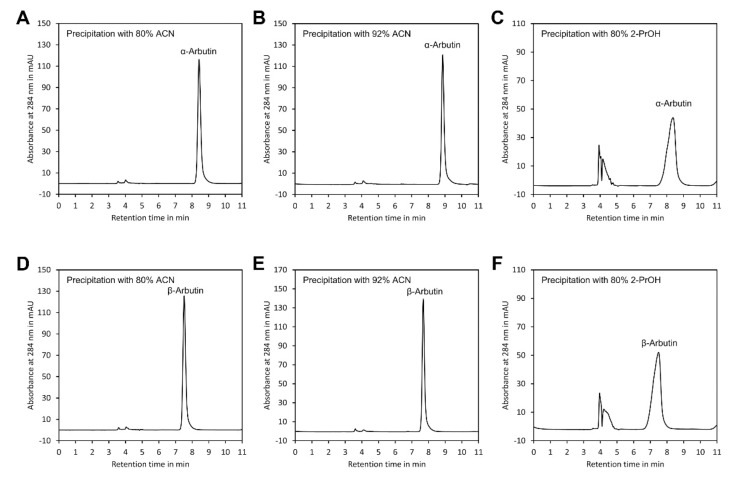
Peak shape of α- and β-arbutin after precipitation of polymers. Peak shape of α-arbutin after precipitation with: (**A**) ACN at a final ratio of ACN/water 80/20 (*v*/*v*), (**B**) ACN/water 92/8 (*v*/*v*) and (**C**) 2-PrOH/water 80/20 (*v*/*v*). Peak shape of β-arbutin after precipitation with: (**D**) ACN at a final ratio of ACN/water 80/20 (*v*/*v*), (**E**) ACN/water 92/8 (*v*/*v*) and (**F**) 2-PrOH/water 80/20 (*v*/*v*). The original data are available in Appendix A.

**Figure 5 molecules-27-05673-f005:**
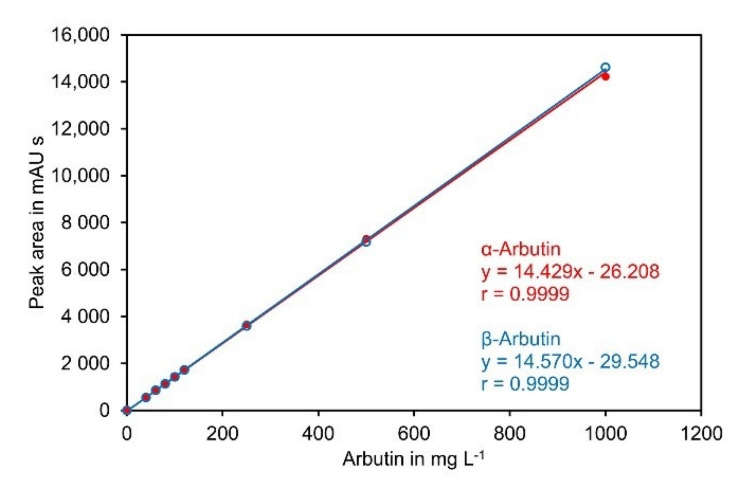
Calibration curve for α-arbutin and β-arbutin. Standards containing both compounds at the indicated concentrations were separated on a Cyclobond I 2000 5 µm 250 × 4.6 mm column using ACN/water 92/8 (*v*/*v*) at a flow rate of 0.8 mL min^−1^ as eluent. UV absorbance was measured at 284 nm. The original data are available in Appendix A.

**Figure 6 molecules-27-05673-f006:**
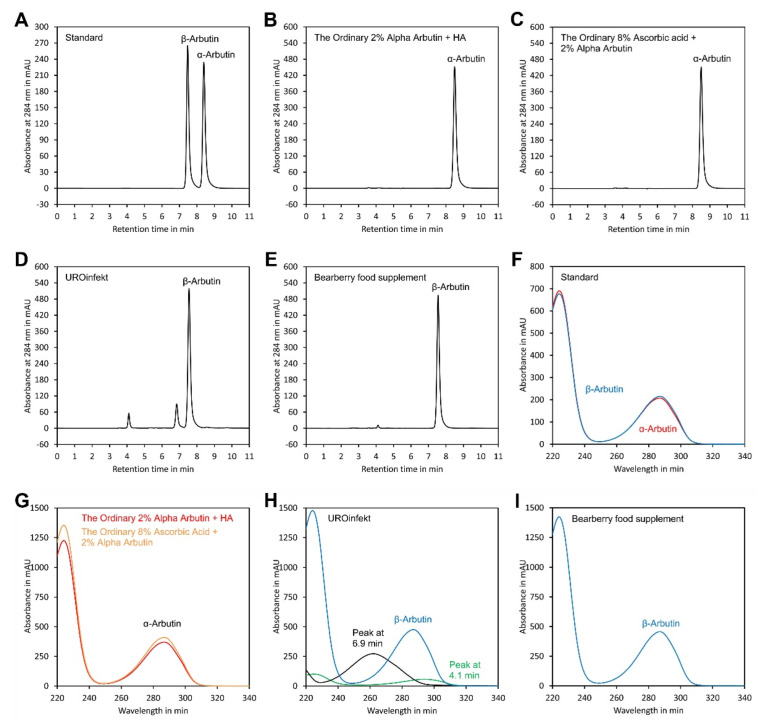
Analysis of samples. Chromatograms of (**A**) a standard containing authentic α-arbutin and β-arbutin; and samples of (**B**) The Ordinary Alpha Arbutin 2% + HA, (**C**) The Ordinary 8% Ascorbic Acid + 2% Alpha Arbutin, (**D**) UROinfekt for treatment of lower urinary tract infections and (**E**) bearberry extract food supplement. (**F**) Spectra of the α-arbutin and β-arbutin peaks of the standard. (**G**) Overlay of the spectra of the α-arbutin peaks from samples shown in (**B**,**C**). (**H**) Spectra of the β-arbutin peak and the peaks at 4.1 and 6.9 min from the sample shown in (**D**). (**I**) Spectra of the β-arbutin peak from the sample shown in (**E**). The original data are available in Appendix A.

**Table 1 molecules-27-05673-t001:** Recovery rates and peak width at half peak height of α- and β-arbutin ^1^.

Solvent, Final Content (*v*/*v*)	α-Arbutin Recovery ± SD ^2^ in %	α-Arbutin W_1/2_ in Min	β-Arbutin Recovery ± SD in%	β-Arbutin W_1/2_ in Min
ACN/H_2_O 80/20	100.8 ± 2.0	0.21	103.6 ± 2.5	0.19
ACN/H_2_O 92/8	101.4 ± 1.7	0.18	102.2 ± 0.9	0.16
2-PrOH/H_2_O 80/20	101.3 ± 2.0	0.53	101.1 ± 2.5	0.49

^1^ The original data are available in Appendix A. ^2^ SD, standard deviation.

**Table 2 molecules-27-05673-t002:** Recovery rates of α- and β-arbutin in different matrices ^1^.

Matrix ^2^	Analyte	Unspiked	Spiked	Added	Found	Recovery Rate
Matrix 1	α-Arbutin	1.99 ± 0.03% (*w*/*w*)	4.04 ± 0.01% (*w*/*w*)	2% (*w*/*w*)	2.05 ± 0.04% (*w*/*w*)	102.5 ± 1.8%
Matrix 2	α-Arbutin	2.01 ± 0.03% (*w*/*w*)	4.08 ± 0.02% (*w*/*w*)	2% (*w*/*w*)	2.07 ± 0.03% (*w*/*w*)	103.4 ± 1.7%
Matrix 3	β-Arbutin	15.18 ± 0.34 g L^−1^	35.62 ± 0.42 g L^−1^	20 g L^−1^	20.44 ± 0.51 g L^−1^	102.2 ± 2.6%
Matrix 4	β-Arbutin	10.66 ± 0.22 g L^−1^	31.09 ± 0.32 g L^−1^	20 g L^−1^	20.43 ± 0.47 g L^−1^	102.2 ± 2.4%

^1^ The original data are available in Appendix A. ^2^ Matrix 1: The Ordinary Alpha Arbutin 2% + HA; matrix 2: The Ordinary Ascorbic Acid 8% + Alpha Arbutin 2%; matrix 3: solution of UROinfekt; matrix 4: solution of bearberry food supplement.

**Table 3 molecules-27-05673-t003:** Intra-day and inter-day repeatability α-arbutin in The Ordinary 2% Arbutin.

Experiment ^1^	Replicates	α-Arbutin ^2^ Average ± SD (RSD) % (*w*/*w*) (%)
Day 1	4	1.910 ± 0.029 (1.5)
Day 2	4	1.908 ± 0.025 (1.3)
Day 3	4	1.910 ± 0.010 (0.5)
Day 4	4	1.902 ± 0.018 (0.9)
Inter-day	16	1.907 ± 0.020 (1.0)

^1^ The original data are available in Appendix A. ^2^ β-Arbutin was below the detection limit.

**Table 4 molecules-27-05673-t004:** Intra-day and inter-day repeatability α-arbutin in The Ordinary 8% Ascorbic Acid + 2% Arbutin.

Experiment ^1^	Replicates	α-Arbutin ^2^ Average ± SD (RSD) % (*w*/*w*) (%)
Day 1	3 ^3^	1.879 ± 0.043 (2.3)
Day 2	4	1.909 ± 0.038 (2.0)
Day 3	4	1.898 ± 0.009 (0.5)
Day 4	4	1.884 ± 0.014 (0.7)
Inter-day	15	1.893 ± 0.028 (1.5)

^1^ The original data are available in Appendix A. ^2^ β-Arbutin was below the detection limit. ^3^ One value was excluded as outliner based on the Grubbs test.

**Table 5 molecules-27-05673-t005:** Intra-day and inter-day repeatability α-arbutin in UROinfekt.

Experiment ^1^	Replicates	β-Arbutin ^2^ Average ± SD (RSD) mg per Pill (%)
Day 1	4	156.2 ± 1.7 (1.1)
Day 2	4	157.2 ± 1.8 (1.2)
Day 3	4	163.5 ± 1.1 (0.7)
Day 4	4	161.1 ± 2.9 (1.8)
Inter-day	16	159.5 ± 3.6 (2.2)

^1^ The original data are available in Appendix A. ^2^ α-Arbutin was below the detection limit.

**Table 6 molecules-27-05673-t006:** Intra-day and inter-day repeatability α-arbutin in bearberry extract food supplement.

Experiment ^1^	Replicates	β-Arbutin ^2^ Average ± SD (RSD) mg per Pill (%)
Day 1	4	103.1 ± 0.6 (0.6)
Day 2	4	104.2 ± 1.1 (1.1)
Day 3	4	104.6 ± 0.2 (0.2)
Day 4	4	103.1 ± 1.0 (1.0)
Inter-day	16	103.8 ± 1.0 (1.0)

^1^ The original data are available in Appendix A. ^2^ α-Arbutin was below the detection limit.

**Table 7 molecules-27-05673-t007:** Calibration curve equations and Pearson correlation coefficients.

Experiment ^1^	α-Arbutin Equation (r) ^2^	β-Arbutin Equation (r)
Day 1	y = 14.452x − 17.940 (0.9999)	y = 14.669x − 34.583 (0.9999)
Day 2	y = 14.429x − 26.208 (0.9999)	y = 14.570x − 29.548 (0.9999)
Day 3	y = 14.353x − 21.773 (0.9999)	y = 14.579x − 38.618 (0.9998)
Day 4	y = 14.254x − 4.355 (0.9997)	y = 14.483x − 7.216 (0.9998)

^1^ The original data are available in Appendix A. ^2^ r, Pearson correlation coefficient.

## Data Availability

The raw data obtained in this study are summarised in Appendix A.

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
