# Peer review of "Quantification of Arbutin in Cosmetics, Drugs and Food Supplements by Hydrophilic-Interaction Chromatography"

_molecules, 2022, doi:10.3390/molecules27175673_

Round 1
Reviewer 1 Report (Previous Reviewer 2)
This manuscript presents the development of a new HILIC approach for the analysis of arbutin isomers in cosmetics, drugs and food supplements. The introduction is well documented with up-to-date references. The methods are described adequately, while the results and discussion are thorough. I consider that this manuscript can be accepted for publication.
Several minor comments:
Abstract: the description of the analysis protocol should be made at past tense.
Lines 78-81: this part needs rephrasing, since some techniques involve separation before detection (TLC, CZE, MECK and GC), while spectrophotometry does not; on the other hand electrochemistry is a whole science field, therefore a more precise description should be provided.
Line 180: "Outliners" should read "outliers"
Figure 1B and C and Line 198: the figures state that the lowest pH is 2, while the caption says 1. Please correct.
Figure 5: figure should be removed. Calibration curve can be described by the regression equations (already provided in Table 7), while calibration levels should be provided in the text.
Author Response
Reviewer 1
This manuscript presents the development of a new HILIC approach for the analysis of arbutin isomers in cosmetics, drugs and food supplements. The introduction is well documented with up-to-date references. The methods are described adequately, while the results and discussion are thorough. I consider that this manuscript can be accepted for publication.
We want to thank reviewer 1 for positive evaluation and helpful comments. We have given each point careful consideration as described below. The original text of reviewer 1 is in black while our answers are in blue. Changes in the manuscript are marked using the track changes function.
Several minor comments:
Abstract: the description of the analysis protocol should be made at past tense.
Answer 1
Done as suggested.
Lines 78-81: this part needs rephrasing, since some techniques involve separation before detection (TLC, CZE, MECK and GC), while spectrophotometry does not; on the other hand electrochemistry is a whole science field, therefore a more precise description should be provided.
Answer 2
Many thanks for this comment. We agree that separation techniques should be discussed separately from spectroscopic methods. The text was adapted accordingly.
Line 180: "Outliners" should read "outliers"
Answer 3
Corrected as suggested.
Figure 1B and C and Line 198: the figures state that the lowest pH is 2, while the caption says 1. Please correct.
Answer 4
Many thanks for pointing that out. The caption and running text were corrected.
Figure 5: figure should be removed. Calibration curve can be described by the regression equations (already provided in Table 7), while calibration levels should be provided in the text.
Answer 5
We agree that showing the calibration curve is not essential. However, with this graph we want to illustrate the almost identical detector response for both compounds (see line 238). Thus, we would prefer to keep this figure.

Reviewer 2 Report (New Reviewer)
Manuscript ID molecules-1887103 authored by Dr. Sarah Repert et al is an interesting research on the development and validation of an HILIC method for the quantification of both alpha and beta arbutin. The method was applied on real samples (cosmetic products, a drug and a food supplement). The manuscript seems well written and documented. There are some issues that may improve the value of this manuscript:
1. Row 32 - Please replace "skin bleaching" with a must suitable word, maybe whitening.
2. Rows 63-65. This phrase states some information that are not quite updated. Please study the report: https://health.ec.europa.eu/publications/safety-alpha-arbutin-and-beta-arbutin-cosmetic-products_en
3. Rows 68 and 70 - Please use EMA instead of EMEA
4. Rows 78-87 - Please use updated references
5. Rows 87-89 - I was nor able to trace the information in reference 22.
6. Section 2.2 - Please describe the cuvettes for UV spectra measurements and also describe the polarimeter in the proper way (Company, City, Country)
7. Section 2.3 - Please describe the centrifuge (Company, City, Country)
8. Please use some other term for "recovery rate" allover the manuscript
9. Row 171 - please mention declared arbutin content in mg per pill
10. Table 2 is misleading . From these data we can conclude that there is no arbutin in none of the samples (Matrix 1 - Matrix 4)
11. Please use the proper name of the drug (UROinfekt) allover the manuscript
12. Table 2 is misleading . From these data we can conclude that there is no arbutin in none of the samples (Matrix 1 - Matrix 4)
13. The term "Recovery" use inside Tables 3, 4, 5 and 6 is not appropriate
14. I am concerned about the small amounts of arbutin stated in Table 5. As far as I understand declared content is 200 mg/tablet with a range of 180-210 mg/tablet. You provided the proper explanation (rows 334-340). This is just a thought and you are not asked to do anything.
15. Please update References section according to the rules stated under Molecules journal webpage.
16. Template paper presented under Molecules website has the following sections (in this order): Introduction, Results, Discussion, Materials and Methods, Conclusions..... If you consider proper, please update the manuscript accordingly.
Author Response
Reviewer 2
Manuscript ID molecules-1887103 authored by Dr. Sarah Repert et al is an interesting research on the development and validation of an HILIC method for the quantification of both alpha and beta arbutin. The method was applied on real samples (cosmetic products, a drug and a food supplement). The manuscript seems well written and documented. There are some issues that may improve the value of this manuscript:
We want to thank reviewer 2 for positive evaluation and helpful comments. We have given each point careful consideration as described below. The original text of reviewer 1 is in black while our answers are in blue. Changes in the manuscript are marked using the track changes function.
- Row 32 - Please replace "skin bleaching" with a must suitable word, maybe whitening.
Answer 1
Was corrected as suggested.
- Rows 63-65. This phrase states some information that are not quite updated. Please study the report: https://health.ec.europa.eu/publications/safety-alpha-arbutin-and-beta-arbutin-cosmetic-products_en
Answer 2
Many thanks for this very recent reference. This illustrates the ongoing debate about safety of arbutin very well. We have included this reference and adapted the manuscript accordingly.
- Rows 68 and 70 - Please use EMA instead of EMEA
Answer 3
Corrected as suggested.
- Rows 78-87 - Please use updated references
Answer 4
The paragraph was modified as suggested by reviewer 1 and the references were updated.
- Rows 87-89 - I was nor able to trace the information in reference 22.
Answer 5
In reference 22 (Lamien-Meda et al., 2008; PMID 19609883) arbutin was derivatised using N,O-bis(trimethylsilyl)acetamide and trimethylchlorosilane as silylation reagents. The reaction product was separated by gas chromatography on a DB-5 narrow bore column and detected using a flame ionisation detector or a mass spectrometer. Salicin was used as an internal standard. Using this method arbutin in Origanum majorana and Arctostaphylos uva-ursi were quantified. Since our manuscript focusses on liquid chromatography these details were not included in the introduction.
- Section 2.2 - Please describe the cuvettes for UV spectra measurements and also describe the polarimeter in the proper way (Company, City, Country)
Answer 6
The missing information were added.
- Section 2.3 - Please describe the centrifuge (Company, City, Country)
Answer 7
The missing information were added.
- Please use some other term for "recovery rate" allover the manuscript
Answer 8
The term “recovery rate” is widely used for spiking experiments. Thousands of examples can be found in the literature and there are also lots of manuscripts published in recent volumes of Molecules that use this term. Here are just a few examples:
Wu et al., 2022, https://www.mdpi.com/1420-3049/27/15/4689
Zheng et al., 2022, https://www.mdpi.com/1420-3049/27/13/4087
Zhao et al., 2022, https://www.mdpi.com/1420-3049/27/10/3189
Kast et al., 2022, https://www.mdpi.com/1420-3049/27/17/5367
Shi et al., 2022, https://www.mdpi.com/1420-3049/27/7/2092
Kalogiouri et al., 2022, https://www.mdpi.com/1420-3049/27/4/1435
Thus, since the term “recovery rate” is widely used for evaluation of chromatographic and other analytical methods, we kept this term in our manuscript.
- Row 171 - please mention declared arbutin content in mg per pill
Answer 9
The declared β-arbutin content was included in the text.
- Table 2 is misleading . From these data we can conclude that there is no arbutin in none of the samples (Matrix 1 - Matrix 4)
Answer 10
We agree that Table 2 was misleading and thus we have revised this table completely. Now the amount before spiking, after spiking, added and found are indicated in Table 2 (previously these data were only shown in Suppl. Table 8).
- Please use the proper name of the drug (UROinfekt) allover the manuscript
Answer 11
The name of the drug was corrected in the whole manuscript.
- Table 2 is misleading . From these data we can conclude that there is no arbutin in none of the samples (Matrix 1 - Matrix 4)
Answer 12
Please see question and answer 10.
- The term "Recovery" use inside Tables 3, 4, 5 and 6 is not appropriate
Answer 13
Many thanks for pointing that out! The correct term is of course average. The tables were corrected accordingly.
- I am concerned about the small amounts of arbutin stated in Table 5. As far as I understand declared content is 200 mg/tablet with a range of 180-210 mg/tablet. You provided the proper explanation (rows 334-340). This is just a thought and you are not asked to do anything.
Answer 14
We were also surprised about that low level and thus check the product description in detail, where the analytical method (UV spectroscopy) was mentioned. With think that the lower specificity accounts for overestimation of the arbutin content in this product, as we have discussed in the text.
- Please update References section according to the rules stated under Molecules journal webpage.
Answer 15
The References section was updated according to the rules.
- Template paper presented under Molecules website has the following sections (in this order): Introduction, Results, Discussion, Materials and Methods, Conclusions..... If you consider proper, please update the manuscript accordingly.
Answer 16
We have obviously used an old template where the order was different. We have updated the order of the sections. This was done without “track changes” to prevent that the revised manuscript becomes confusing.

This manuscript is a resubmission of an earlier submission. The following is a list of the peer review reports and author responses from that submission.
Round 1
Reviewer 1 Report
To authors:
· The Abstract is meaningful.
· The introduction provides sufficient background understanding. However some concepts presented are not very clear to the reader. Please see my suggested comments.
· The references are appropriate.
· Presentation of data/information: Please see my comments/suggestions listed below.
· Figures: Acceptable.
· Overall, this manuscript is missing many fundamental aspects of a quality manuscript. Therefore, this manuscript does not meet publishing criteria of the journal.
1. Line 77-79: References? Usually isomers can be separated by RPLC if the method is well developed.
2. Line 82-84: Are you referring to matrix effect? On C18 or RPLC arbutin will elute way earlier than matrix molecules, let's say PEG. Having organic placebo molecules eluting later and having broader peak shape would not affect quantification of arbutin. I suggest to reword this statement because it is not accurate to generalize this concept.
3. Line 94: What is the reason to choose Cyclobond column? Did you screen other chiral HILIC phases?
4. All over the manuscript: Please specify the mobile phase composition clearly. Example 92% (v/v) ACN, What is the other component? Readers won't assume it as ACN/Water, Did you use buffers/additives?
5.Line 140: VICH is for veterinary medicine, not for human use. For your purpose relevant reference is ICH Q2 and Q3
6. Line 141: Please define SD
7. Line 145: Pease define HA
8. Section 3.1: Recommend to move Optical isomer confirmation section to supporting information, You can use single isomer standard as a RT marker in your method. Polarimetry is just to confirm identity of commercial material you purchased. If you used CD (Chiral detector) in your work, it is worth mentioning.
9.Line 200-202: These results were obtained under optimized conditions? Please compare optimized separations on 3 stationary phases to be fair.
10. Table1. Please provide full description of solvent composition
11.Recovery study: Recovery study for method validation is performed by spiking known amount in to the matrix/placebo. What is presented here is actual sample analysis. Please clarify. To compare your quantification you may need to actual arbutin amount in the product which needs to be provided by the cosmetics manufacturer (in the form of CoA). This 2% is label information (known as Label Claim). It does not mean any actual number.
Reviewer 2 Report
The manuscript describes the development and validation of a HPLC-UV method which aims to analyze arbutin isomers in different cosmetic preparations, drugs and food supplements. The resolution of the two arbutin isomers is achieved by HILIC, which is appropriate considering the highly hydrophilic nature of the analytes. In my opinion this manuscript could be accepted for publication after dealing with the following aspects:
1. There are several aspects in the introduction which need improvement:
a. The literature documented in the first part can be improved with more recent studies. Recently, the Scientific Committee on Consumer Safety of EU released an "Opinion on the safety of alpha-arbutin and beta-arbutin in cosmetic products" (https://ec.europa.eu/health/system/files/2022-03/sccs_o_264.pdf)
b. Line 52: You stated: "The half maximal inhibitory concentration (IC50) of arbutin is 17 mM [7]" – please describe the whole context of that determination (animal, mode of admin)
c. Line 54-55: Reference 8 should contain the original EMA/ EU statement
d. Line 58: the paper in reference 10 describes the development of a method. There is no reporting of testing arbutin irritation on the skin or other side effects. Please replace with the appropriate reference.
e. Since this manuscript describes the development of a new analytical approach, I believe more information on the state-of-the-art in arbutin analysis should be added.
2. Section 2.5 Validation: It seems that the LOD and LOQ estimation was not performed as in annex 2 of the VICH GL49(R) guideline, but rather annex 1. This approach is not wrong, but probably not the most accurate way. On top, in my opinion the VICH GL49(R) is not of reference for the type of analytical method development that is described in the manuscript. A more appropriate guide could be the ICH Q2(R1) (https://ec.europa.eu/health/system/files/2022-03/sccs_o_264.pdf) which offers three rather simple ways to measure or estimate the LOD and LOQ.
3. Other points:
Line 136 and elswere in text: please replace "to retard/retardation" with "to retain/retention" when talking about chromatographic retention of an analyte.
Figure 5: this figure can be omitted, since the linearity of the calibration curves can be explained by an equation.
Reviewer 3 Report
The present manuscript entitled "Quantification of arbutin in cosmetics, drugs and food supplements by hydrophilic-interaction chromatography" by Sarah Repert, Sandra Matthes, and Wilfried Rozhon (molecules-1794852) is written correctly and has a good structure; moreover, it has all the necessary parts. The article is interesting from an analytical point of view; therefore, it should interest the reader. I proposed improvements in method description and with a presentation of figures. The paper meets Molecules' requirements, and I recommend the article for publication in Molecules following the common editing stage. My current decision is a minor revision. More specific comments and observations are presented below.
1. The authors mentioned that method is free from interferences. What can be done in the event of strong interference effects? How would you deal with them? What types of interference effects could occur?
2. RSD expressed as a percentage is the coefficient of variation (CV).
3. Introduction. More details on the chromatographic methods used may be added.
4. Section 2.3. What were the parameters of distilled water?
5. Figures with spectra. You can describe the most important maxima numerically in the figures.
6. Figures with chromatograms. It should be "Time" instead of "Retention Time". This is not a retention time chart.
7. Figures 2 D and 2 E. “alpha” and “beta” should be converted to symbols.
8. Figure 6 should be more commented in the text.
9. Does the developed method have disadvantages? What are the limitations?
10. Conclusion. Please, emphasize clearly the advantages of the research carried out.
11. It would be worthwhile to evaluate the method using RGB Additive Color Model to Analytical Method Evaluation or AGREE-Analytical GREEnness Metric Approach.
I hope that the comments presented will help improve the article.
Round 2
Reviewer 1 Report
Thank you very much for taking time to work on revisions. Please see below listed comments.
Line 142: What is the gradient program? Please provide compositions over time.
Line 149: What is rest of the solvent? Water? Please state clearly. These mobile phase compositions are very confusing to readers. For example if your mobile phase is 92% ACN and 8% water, It can be best written as ACN/Water (92/8, v/v). The way you have presented the compositions, it is not easily readable. This is applicable to entire manuscript.
Section 3. Recovery experiments: If this (SPN-5) is not the exact placebo/matrix present in your tested cosmetic formulations, your experiment is invalid. It does not matter if SPN-5 is free of Arbutin.